# Sub-THz Small-Signal Equivalent Circuit Model and Parameter Extraction for 3 nm Gate-All-Around Nanosheet Transistor

Yabin Sun [1,2], Hengbin Gao [2], Shaojian Hu [3], Ziyu Liu [1,*], Xiaojin Li [2], Yun Liu [2] and Yanling Shi [2]

[1]  School of Microelectronics, Fudan University, Shanghai 200433, China; ybsun@ee.ecnu.edu.cn
[2]  Department of Electrical Engineering, East China Normal University, Shanghai 200241, China; ecnu_ee_ghb@foxmail.com (H.G.); xjli@ee.ecnu.edu.cn (X.L.); yliu@cee.ecnu.edu.cn (Y.L.); ylshi@eeecnu.edu.cn (Y.S.)
[3]  Shanghai Integrated Circuit Research and Development Center, Shanghai 201210, China; shaojianh@icrd.cn
*   Correspondence: liuziyu@fudan.edu.cn

**Abstract:** This paper presents a novel RF small-signal equivalent circuit model and parameter extraction for 3 nm nanosheet gate-all-around field effect transistor (GAAFET). The extrinsic parasitic effect induced by ground-signal-ground (GSG) layout is evaluated by 3D full-wave electromagnetic simulation, and an improved five-step analytical parameter extraction method is proposed for such extrinsic GSG layout. The model parameters for the intrinsic device are analytically determined with the help of nonlinear rational function fitting. The accuracy of the proposed extraction method was confirmed via comparisons between device simulator and electromagnetic simulator with frequency responses up to 300 GHz. Excellent agreement is obtained between the simulated and modeled S-parameters, and the calculated error is lower than 2.689% for the extrinsic layout, and 0.897% for the intrinsic device in the whole frequency range among multi-bias points.

**Keywords:** gate-all-around field effect transistor; small signal equivalent circuit model; parameter extraction; full-wave electromagnetic simulation

## 1. Introduction

As the feature size continues to shrink, traditional planar MOSFETs can no longer meet the scaling rules of Moore's law, suffering from the serious short channel effects [1]. Multi-gate structures such as FinFET have been applied to industrial manufacturing since the 22 nm technology node [2]. When entering to sub 3 nm node, the gate-all-around (GAA) nanosheet architecture has become a convincing candidate beyond FinFET [3]. When entering millimeter-wave even T-Hz band, the transistors will experience severe parasitic effects [4]. An accurate physically oriented circuit model and parameter extraction technique for a GAA nanosheet transistor are very important to evaluate the fabrication process, optimize the device structure and help in circuit design [5–7].

The accurate de-embedding is the first step to model the high-frequency behavior and extract the model parameters. Generally, the OPEN and SHORT method has been widely adopted to strip the parasitism of the test pad and interconnect line over the low frequency range [8]. However, the efficiency and accuracy of the OPEN-SHORT de-embedding method will degrade seriously when the frequency is larger than 40 GHz. Thus, some improved de-embedding techniques such as the OPEN-SHORT-THRU method and multi-step de-embedding structures [9,10]. For higher operation frequency such as mmW or T-Hz range, 3D full-wave electromagnetic field simulation has also been proposed to model the extrinsic parasitism coupling and extract the PAD parasitism, which has been successfully applied in III–V devices and planar MOSFET [10–12]. However, few reports document electromagnetic simulation to extract the extrinsic parasitic effects of FinFET or GAAFET, because the parasitic effects are more diversified suffering the complex three-dimensional ring gate, multi-fingered electrode and substrate contact ring structure.

Over the last three decades, several research studies have focused on the parameter extraction for the MOSFET small-signal equivalent circuit model [13–16]. Earlier works generally rely on numerical optimization to match the simulated curves with the measurement data [14]. However, the extracted parameters are often sensitive to the initial-guess values, and some non-physical results may be obtained such as a negative capacitance or resistance. To overcome this problem, another analytical or semianalytical methods are proposed [13,15,16]. For example, all the signal model parameters of FinFET are analytically extracted through S-parameter under different bias conditions, and the high accuracy is obtained over the 50 GHz frequency range [16]. The substrate $R_{sub}$ including the bulk effect is extracted by shorting the source and drain, which improves the accuracy of $Y_{22}$ at the high frequency range [13]. The gate parasitic resistance $R_g$, which is important for the frequency characteristics, is ignored in several works [17].

In this paper, a novel analytical parameter extraction method is established for RF small-signal equivalent circuit of GAAFET at 3 nm node. The extrinsic parasitic parameters of the 3D test layout are extracted and compared to HFSS 3D full-wave electromagnetic simulation. The 3 nm GAAFET was built and simulated in Sentaurus TCAD to extract intrinsic parasitic parameters. Excellent agreement between the modeled and simulated data illustrates the improved modeling accuracy.

## 2. Material and Methods

### 2.1. Device Structure and Equivalent Circuit Model

The 3D schematic view of the device structure for the vertically stacked GAA NSFET is shown in Figure 1a–c, depicting the 2D cross-section views along the channel and gate direction. The physical gate length $L_g$ is 15 nm. Three nanosheets with ellipse cross-section are vertically stacked. The width and height of each nanosheet are separately 20 nm and 5 nm. The gate oxide is composed of $SiO_2$ and $HfO_2$ and the equivalent oxide thickness is 0.68 nm. The spacer is composed of 4.2 nm $SiO_2$ and 1.6 nm $HfO_2$. The doping of the channel and source/drain are $1 \times 10^{15}$ cm$^{-3}$ and $1 \times 10^{21}$ cm$^{-3}$, respectively. TiN with aa work function of 4.37 eV is adopted as the gate metal. Figure 2 shows the adopted two-finger coplanar GSG PAD layout to de-embed the impact of the test structure and the compounding equivalent circuit model. Each parasitic parameter from the layout is also marked out. Here, $(C_{gde}, G_{gde})$, $(C_{gse}, G_{gse})$ and $(C_{dse}, G_{dse})$ are the interconnection parasitic capacitance–conductance pairs between PAD electrodes. $(R_{ge}, L_{ge})$, $(R_{de}, L_{de})$ and $(R_{se}, L_{se})$ represent the parasitic resistance–inductance pair of PAD electrodes. $(C_{gdp}, G_{gdp})$, $(C_{gsp}, G_{gsp})$ and $(C_{dsp}, G_{dsp})$ are interconnection capacitance–conductance pairs between PAD electrodes.

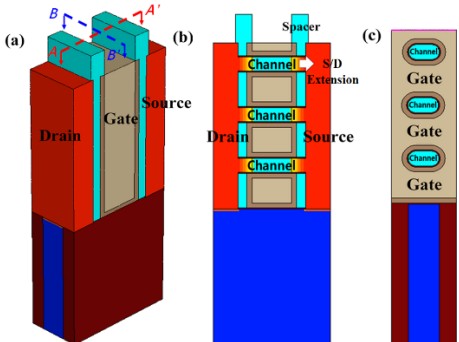

**Figure 1.** Intrinsic 3 nm nanosheet GAAFET. (**a**) 3D schematic and (**b**) 2D cross-section along AA' direction and (**c**) BB' direction.

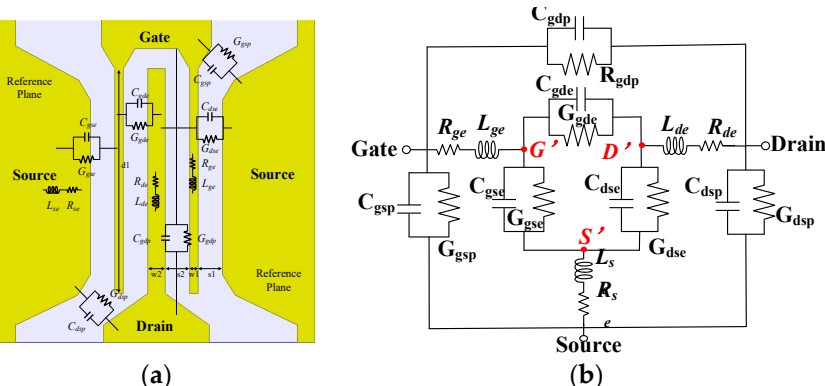

**Figure 2.** (**a**) The adopted two-finger coplanar GSG PAD layout and (**b**) the equivalent circuit of GSG PAD layout.

Figure 3 shows the proposed improved small-signal equivalent circuit model of intrinsic GAAFET. $C_{gdo}$ and $C_{gso}$ are the capacitance between the gate and source/drain. $R_g$, $R_s$, $R_d$ and $R_{sub}$ are separately the resistance of the gate, source, drain and substrate. $C_{jd}$ is the capacitance between the drain and substrate. $C_{gsi}$ and $C_{gdi}$ are the capacitance between the source/drain overlap and gate region. $R_{gsi}$ and $R_{gdi}$ are the resistance of the source and drain overlap region. $R_{ds}$ and $L_{ds}$ are the channel's resistance and inductance. $\tau_m$ characterizes the transmission delay between the source and drain. $C_{sdx}$ characterizes the DIBL effect, and $g_m$ is the transconductance.

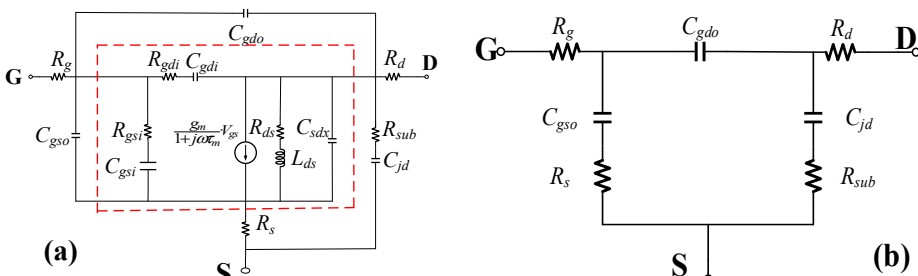

**Figure 3.** The equivalent circuit of intrinsic GAAFET under (**a**) ON-state, where the elements within the redline are bias-dependent, and (**b**) OFF-state, where the elements are bias-independent.

### 2.2. Extrinsic Layout Test Structures and Parameter Extraction Algorithm

As shown in Figure 2b, 18 parasitic elements are contained in the extrinsic GSG layout parasitic circuit. It is unrealistic to extract many parasitic parameters by directly fitting measured or simulated $S$ (or $Y$)-parameters. Thus, a kind of multi-step extraction method has been proposed [10]. However, because the parasitic parameters are sensitive to the layout structure, two kinds of short structures may not match expectations when performing the fourth and fifth steps. Here, we propose an improved multi-step test structure method to better characterize the circuit performance.

First, similar to the conventional method in [18,19], the gate and drain electrodes are removed, and the PAD of the source, gate and drain is simulated to extract the coupling parasitic capacitance and conductance between PADs. The layout and corresponding sub-circuit model are shown in Figure 4a. The $Y$-matrix of such PADs structure is as follows:

$$Y_{PAD} = \begin{pmatrix} G_{gsp} + j\omega C_{gsp} + G_{gdp} + j\omega C_{gdp} & -G_{gdp} - j\omega C_{gdp} \\ -G_{gdp} - j\omega C_{gdp} & G_{dsp} + j\omega C_{dsp} + G_{gdp} + j\omega C_{gdp} \end{pmatrix} \quad (1)$$

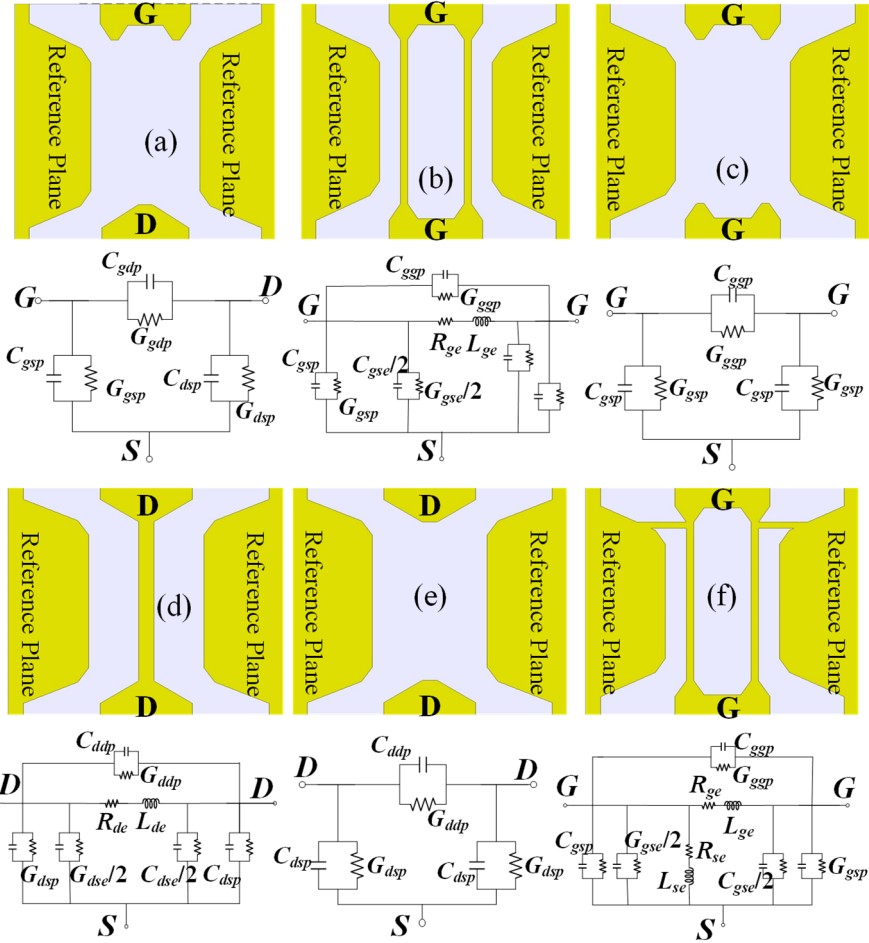

**Figure 4.** Improved test structure and corresponding sub-circuit model for extrinsic G parasitic circuit extraction. (**a**) PAD; (**b**) THRU1; (**c**) THRU1_PAD; (**d**) THRU2; (**e**) THRU2_PAD; and (**f**) THRU1_SHORT.

Thus, $G_{gdp}$, $G_{gsp}$ and $G_{dsp}$ can be obtained from the real part of $Y_{pad,12}$, $Y_{pad,11}$ and $Y_{pad,22}$, and $C_{gdp}$, $C_{gsp}$ and $C_{dsp}$ are the slope of the imaginary part of $Y_{pad,12}$, $Y_{pad,11}$ and $Y_{pad,22}$ verses $\omega$, as depicted below.

$$G_{gdp} = Re\left(-Y_{pad,12}\right) \tag{2}$$

$$G_{gsp} = Re\left(Y_{pad,11}\right) - Re\left(-Y_{pad,12}\right) \tag{3}$$

$$G_{dsp} = Re\left(Y_{pad,22}\right) - Re\left(-Y_{pad,12}\right) \tag{4}$$

$$C_{gdp} = Im\left(-Y_{pad,12}\right)/\omega \tag{5}$$

$$C_{gsp} = \left[Im\left(Y_{pad,11}\right) - Im\left(-Y_{pad,12}\right)\right]/\omega \tag{6}$$

$$C_{dsp} = \left[Im\left(Y_{pad,22}\right) - Im\left(-Y_{pad,12}\right)\right]/\omega \tag{7}$$

In the second step, the THRU1 structure is introduced to extract $R_{ge}$ and $L_{ge}$. The lower half of the drain electrode is removed and the upper half is mirrored to the lower half. Then, the gate electrode is extended to connect the lower PAD, as shown in Figure 4b. Based on such THUI1 structure, the THRU1_PAD structure will be obtained when the gate electrode is removed and the corresponding sub-circuit is depicted, as seen in Figure 4c. After de-embedding the THRU1_PAD from THRU1, the admittance matrix of the gate electrode $Y_{GATE} = Y_{THU1} - Y_{THU1\_PAD}$ is calculated as:

$$Y_{GATE} = \begin{pmatrix} (G_{gse} + j\omega C_{gse})/2 + (R_{ge} + j\omega L_{ge})^{-1} & (G_{gse} + j\omega C_{gse})/2 + (R_{ge} + j\omega L_{ge})^{-1} \\ -(R_{ge} + j\omega L_{ge})^{-1} & -(R_{ge} + j\omega L_{ge})^{-1} \end{pmatrix} \tag{8}$$

Thus, similar to the extraction of $G_{gdp}$, $G_{gsp}$, $G_{dsp}$, $C_{gdp}$, $C_{gsp}$ and $C_{dsp}$ in Equations (2)–(7), the gate electrode resistance and inductance $R_{ge}$ and $L_{ge}$ can also be easily obtained.

In the third step, a kind of THRU2 structure is introduced. The upper half of the gate electrode is removed, and the lower half is mirrored to the upper half, then the drain electrode is extended to connect to the upper and lower PAD, as shown in Figure 4d. The corresponding THRU2_PAD structure is depicted in Figure 4e by removing the drain electrode in the THUR2 structure. The corresponding sub-circuits are separately depicted in Figure 4d,e. Similarly, the admittance matrix of the drain electrode $Y_{DRAIN} = Y_{THU2} - Y_{THU2\_PAD}$ is written as:

$$Y_{DRAIN} = \begin{pmatrix} (G_{dse} + j\omega C_{dse})/2 + (R_{de} + j\omega L_{de})^{-1} & -(R_{de} + j\omega L_{de})^{-1} \\ -(R_{de} + j\omega L_{de})^{-1} & (G_{dse} + j\omega C_{dse})/2 + (R_{de} + j\omega L_{de})^{-1} \end{pmatrix} \tag{9}$$

So far, the drain electrode resistance $R_{de}$ and inductance $L_{de}$ are obtained.

In the fourth step, a kind of THRU1_SHORT is structured to extract $R_{se}$ and $L_{se}$. The gate electrode is connected to the source PAD based on the THRU1 structure, as shown in Figure 4f. The $Y$-matrix of the source electrode $Y_{SOURCE}$ is calculated as:

$$Y_{SOURCE} = \begin{pmatrix} 0 & -(R_{se} + j\omega L_{se})^{-1} \\ -(R_{se} + j\omega L_{se})^{-1} & 0 \end{pmatrix} \tag{10}$$

Thus, the source electrode resistance $R_{se}$ and inductance $L_{se}$ can be easily obtained.

In the fifth step, the entire layout test structure, as shown in Figure 2a, is simulated to obtain the remaining inter-electrode parasitic capacitance and inductance pairs ($C_{gde}$, $G_{gde}$), ($C_{gse}$, $G_{gse}$) and ($C_{dse}$, $G_{dse}$). The corresponding circuit is shown in Figure 2b. The parasitic network on the electrodes can be expressed as:

$$Z_{SERIES} = \begin{pmatrix} (R_{ge} + j\omega L_{ge}) + (R_{se} + j\omega L_{se}) & R_{se} + j\omega L_{se} \\ R_{se} + j\omega L_{se} & (R_{de} + j\omega L_{de}) + (R_{se} + j\omega L_{se}) \end{pmatrix} \tag{11}$$

The $Y$-matrix of the parasitic network between the electrodes can be expressed as:

$$Y_{ELCTR} = \begin{pmatrix} G_{gse} + j\omega C_{gse} + G_{gde} + C_{gde} & -\left(G_{gde} + j\omega C_{gde}\right) \\ -\left(G_{gde} + j\omega C_{gde}\right) & G_{dse} + j\omega C_{dse} + G_{gde} + C_{gde} \end{pmatrix} \tag{12}$$

Similar to extracting the parameters of the PAD structure, all parasitic capacitance and inductance pairs between the electrodes can be extracted here. HFSS simulation is then performed to obtain the high-frequency characteristics of the designed test structure in Figures 2 and 4, and the frequency range is up to 300GHz. Then, all the model parameters related with GSG PAD can be extracted based on Equations (1)–(12).

### 2.3. Improved Extraction Method of Intrinsic Device Equivalent Circuit

As depicted in Figure 3a, the equivalent circuit of the intrinsic device includes 16 circuit elements. In our previous work [5], a two-step method has been proposed to extract intrinsic parameters based on the ON-state and OFF-state. However, gate resistance $R_g$ is ignored, which is very important for the impedance matching and thermal noise of the gate metal [17]. Here, an improved extraction method including gate resistance $R_g$ for intrinsic devices is proposed in this work.

When the device is biased at the OFF-state ($V_{gs} \leq 0$, $V_{ds} = 0$), the equivalent circuit of the intrinsic device will be reduced as shown in Figure 3b. Thus, $Y$-parameters $Y_{11}^o$ of the OFF-state equivalent circuit in Figure 3b can be calculated as follows:

$$Re[Y_{11}^o] = \left( N_{10}\omega^2 + N_{11}\omega^4 + N_{12}\omega^6 \right) / \left( 1 + M_{11}\omega^2 + M_{12}\omega^4 + M_{13}\omega^6 \right) \tag{13}$$

$$Re[Y_{12}^o] = -\left( N_{20}\omega^2 + N_{21}\omega^4 + N_{22}\omega^6 \right) / \left( 1 + M_{11}\omega^2 + M_{12}\omega^4 + M_{13}\omega^6 \right) \tag{14}$$

$$Im[Y_{12}^o] = -\left( N_{30}\omega + N_{31}\omega^3 + N_{32}\omega^5 \right) / \left( 1 + M_{11}\omega^2 + M_{12}\omega^4 + M_{13}\omega^6 \right) \tag{15}$$

$$Re[Y_{22}^o] = \left( N_{40}\omega^2 + N_{41}\omega^4 + N_{42}\omega^6 \right) / \left( 1 + M_{11}\omega^2 + M_{12}\omega^4 + M_{13}\omega^6 \right) \tag{16}$$

$$Im[Y_{22}^o] = \left( N_{50}\omega + N_{51}\omega^3 + N_{52}\omega^5 \right) / \left( 1 + M_{11}\omega^2 + M_{12}\omega^4 + M_{13}\omega^6 \right) \tag{17}$$

The coefficients $N_{ij}$ and $M_{ij}$ are functions of model parameters in Figure 3b, where the coefficients $N_{10}$–$N_{50}$ in the numerator are shown as follows:

$$N_{10} = C_{gdo}^2 R_d + C_{gso}^2 R_s + R_g \left( C_{gdo} + C_{gdo} \right)^2 \tag{18}$$

$$N_{20} = C_{gdo}^2 \left( R_d + R_g \right) + C_{gdo} \left( C_{gso} R_g + C_{jd} R_d \right) \tag{19}$$

$$N_{30} = C_{gdo} \tag{20}$$

$$N_{40} = C_{gdo}^2 \left( R_d + R_g \right) + C_{jd}^2 \left( R_d + R_{sub} \right) + 2C_{gdo} C_{jd} R_d \tag{21}$$

$$N_{50} = C_{gdo} + C_{jd} \tag{22}$$

Since the source and drain are symmetrical, there exists $C_{gdo} = C_{gso}$ and $R_s = R_d$. Thus, the bias-independent parameters ($C_{gso}$, $C_{gdo}$, $R_s$, $R_d$, $R_g$, $C_{jd}$, $R_{sub}$) can be directly obtained by grouping Equations (18)–(22), as shown below:

$$C_{gdo} = C_{gso} = N_{30} \tag{23}$$

$$R_d = R_s = (N_{10} - 2N_{20}) / \left( 2N_{30}^2 - 2N_{50}N_{30} \right) \tag{24}$$

$$R_g = (2N_{20}N_{30} - N_{10}N_{50}) / \left[ 4N_{30}^2 (N_{30} - N_{50}) \right] \tag{25}$$

$$C_{jd} = N_{50} - N_{30} \tag{26}$$

$$R_{sub} = \left[ N_{40} - \left( 2C_{gdo} + C_{jd} \right) C_{jd} R_d - C_{gdo}^2 \left( R_d + R_g \right) \right] / C_{jd}^2 \tag{27}$$

After removing the bias-independent parameter in Figure 3b, the $Y$-parameter of the equivalent circuit within the redline in Figure 3a can be obtained as:

$$Im\left[ Y_{11}^i + Y_{12}^i \right] = K_{10}\omega / \left( 1 + K_{11}\omega^2 \right) \ \left( \text{where } K_{10} = C_{gsi} \right) \tag{28}$$

$$Re\left[ Y_{11}^i + Y_{12}^i \right] = K_{20}\omega^2 / \left( 1 + K_{11}\omega^2 \right) \left( \text{where } K_{20} = C_{gsi}{}^2 R_{gsi} \right) \tag{29}$$

$$Im[-Y_{12}] = K_{30}\omega / \left( 1 + K_{31}\omega^2 \right) \ \left( \text{where } K_{30} = C_{gdi} \right) \tag{30}$$

$$Re[-Y_{12}] = K_{40}\omega^2 / \left( 1 + K_{31}\omega^2 \right) \ \left( \text{where } K_{40} = C_{gdi}{}^2 R_{gdi} \right) \tag{31}$$

$$Im\left[ Y_{21}^i - Y_{12}^i \right] = K_{50}\omega / \left( 1 + K_{51}\omega^2 \right) \ \left( \text{where } K_{50} = -g_m \tau_m \right) \tag{32}$$

$$Im\left[ Y_{12}^i + Y_{22}^i \right] = \left( K_{70}\omega + K_{71}\omega^3 \right) / \left( 1 + K_{72}\omega^2 \right) \ \left( \text{where } K_{70} = C_{sdx} - L_{ds} / R_{ds}^2 \right) \tag{33}$$

$$1/Re\left[Y_{12}^i + Y_{22}^i\right] = K_{80} + K_{90}\omega^2 \left(\text{where } K_{80} = R_{ds}, K_{90} = L_{ds}^2/R_{ds}\right) \tag{34}$$

Then, the bias-dependent parameters ($C_{gsi}$, $C_{gdi}$, $R_{gsi}$, $R_{gdi}$, $\tau_m$, $g_m$, $R_{ds}$, $L_{ds}$, $C_{sdx}$) can be directly obtained by grouping Equations (28)–(34), as shown below:

$$C_{gsi} = K_{10} \qquad C_{gdi} = K_{30} \tag{35}$$

$$R_{gsi} = K_{20}/K_{10}^2 \qquad\qquad R_{gdi} = K_{40}/K_{30}^2 \tag{36}$$

$$\tau_m = -K_{50}/K_{60} \qquad\qquad g_m = K_{60} \tag{37}$$

$$R_{ds} = K_{80} \qquad\qquad L_{ds} = \sqrt{K_{80} \cdot K_{90}} \tag{38}$$

$$C_{sdx} = N_{70} + \sqrt{K_{80} \cdot K_{90}}/K_{80}^2 \tag{39}$$

So far, the model parameters for the intrinsic circuit have been analytically extracted, once the coefficients $N_{10}$–$N_{50}$ and $K_{10}$–$K_{90}$ are accurate.

## 3. Result and Discussion

### 3.1. Validation of Extraction Method for Extrinsic Layout

After accurately extracting the parasitic parameters of the extrinsic GSG test layout, the full-wave EM simulation in HFSS is compared with the ADS simulation using the equivalent circuit. The accuracy of the proposed improved extrinsic layout extraction method is confirmed over 10 MHz–300 GHz. Figure 5 shows the comparison between simulated and modeled *S*-parameters for the standard OPEN test structure in Figure 2a. The S-parameters obtained by the full-wave EM simulation show an excellent agreement with the extracted equivalent circuit over a wide bandwidth, which confirms the accuracy of the proposed improved extraction method.

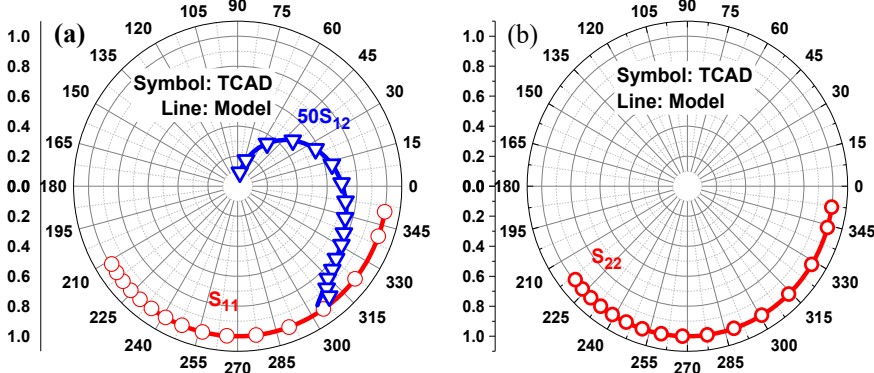

**Figure 5.** Comparison between simulated and modeled S-parameters for the extrinsic layout OPEN structure. (**a**) S$_{11}$, S$_{12}$ and (**b**) S$_{22}$.

Figure 6a shows the extracted inter-electrode and inter-pad capacitors. Consistent with [18,19], $C_{gdp}$ and $C_{gde}$ are much smaller than the other capacitances because of the large space and small co-planar capacitance between the gate and drain electrode [20,21]. Moreover, $C_{gse}$ and $C_{dse}$ decrease with increasing frequency, due to the neglection of the inductance of the source, gate and drain PADs [22]. Figure 6b shows the variation of $G_{gsp}$, $G_{dsp}$, $G_{gdp}$, $G_{gse}$, $G_{dse}$ and $G_{gde}$ as functions of frequencies. For the substrate capacitance and inductance, there is a relation of $G(\omega)/C(\omega) = \omega\tan\delta$ (where $\delta$ represent dielectric constant), and the inductances are basically proportional to its corresponding capacitances [23]. The parasitic resistance–inductance pairs ($R_{ge}$, $L_{ge}$), ($R_{de}$, $L_{de}$) and ($R_{se}$, $L_{se}$) between the source and gate electrodes are shown in Figure 7, respectively. Due to the current crowding phenomenon and the skin effect, the source resistance $R_{se}$ rises rapidly as the frequency enters the mmW band. Due to the skin effect, $L_{ge}$ and $L_{de}$ gradually decrease as the frequency increases. The conductor inductance of the microstrip line is inversely proportional to its

width, while the width of the source electrode is wider than the other electrodes, which leads to a smaller source electrode inductance $L_{se}$.

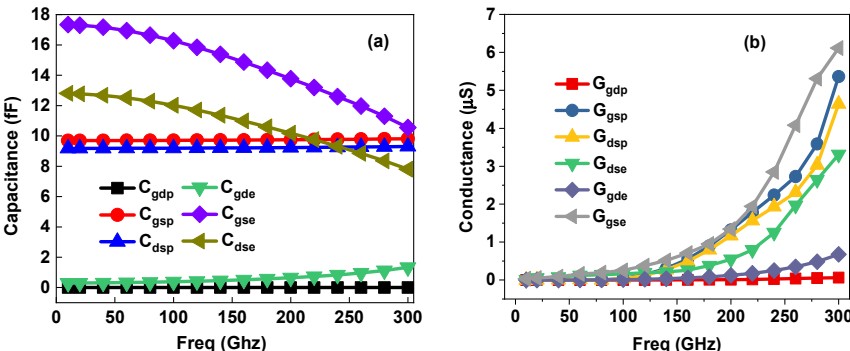

**Figure 6.** (**a**) Inter-electrode equivalent capacitances and (**b**) inter-electrode equivalent conductance of the GSG test layout.

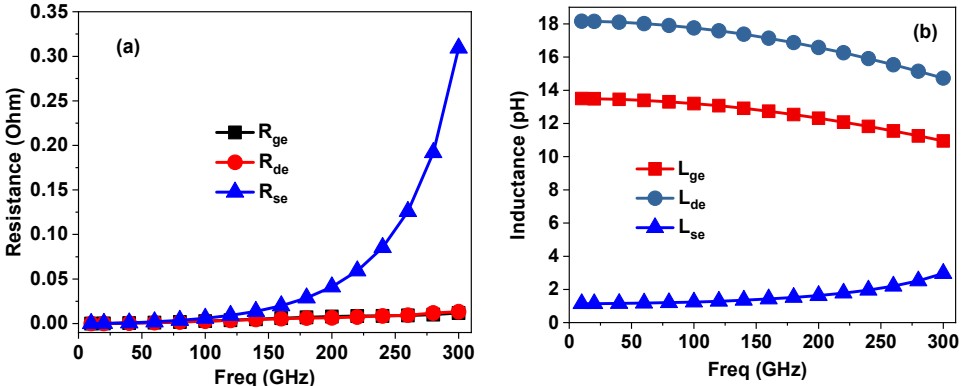

**Figure 7.** The resistance and inductance on the source and drain gate electrodes: (**a**) $R_{ge}$, $R_{de}$ and $R_{se}$; (**b**) $L_{ge}$, $L_{de}$ and $L_{se}$.

### 3.2. Validation of Extraction Method for Intrinsic Device

Sentaurus TCAD is adopted to obtain the high-frequency characteristics of intrinsic NSFET under the ON-state. The critical physical model used in the device simulation includes the drift-diffusion model, density-gradient model, Philips unified mobility model, interface mobility degradation model, velocity saturation and doping-dependent SRH recombination. The simulated high-frequency S-parameters are used to verify the accuracy and feasibility of the small-signal circuit and parameter extraction method. With the help of mathematical fitting software such as MATLAB, the lowest order terms $N_{i0}$ and $K_{i0}$ in Equations (12)–(16) and (22)–(29) can be accurately determined. As for the 3 nm nanosheet transistor in the proposed *OFF*-state bias ($V_{gs} = -0.65$ V, $V_{ds} = 0$ V), the fitting results of $N_{i0}$ and the corresponding confidence interval are shown in Table 1. The narrow confidence interval verifies the accuracy of $N_{i0}$. According to Equations (17)−(21), the bias-independent parameters under such OFF-state are separately extracted as: $C_{gso} = C_{gdo} = 1.466 \times 10^{-17}$ F, $R_s = R_d = 837.16$ Ω, $R_g = 17.29$ Ω, $C_{jd} = 9.9 \times 10^{-19}$ F and $R_{sub} = 9943.3$ Ω. The comparison between the TCAD device simulation in Sentaurus and equivalent circuit simulation in ADS is shown in Figure 8, from which a good agreement exists over 10 MHz–300 GHz.

Similar to the above OFF-state, the coefficients $K_{i0}$ for the ON-state can also be accurately determined. For example, the fitting results for the linear region ($V_{gs} = 0.65$ V, $V_{ds} = 0.05$ V) are shown in Table 2. The reliability of $K_{i0}$ can also be proved by the corresponding narrow confidence interval. According to Equations (30)–(34), the bias-independent intrinsic model parameters under such ON-state are calculated as: $C_{gsi} = 2.968 \times 10^{-17}$ F, $C_{gdi} = 2.77 \times 10^{-17}$ F, $R_{gsi} = 916.33$ Ω, $R_{gdi} = 1331.3$ Ω, $\tau_m = 32.2$ fs, $g_m = 2.71 \times 10^{-5}$ S, $C_{sdx} = 1.90 \times 10^{-18}$ F and $R_{ds} = 2357$ Ω. S-parameter comparison between the Sentaurus device simulation and ADS

equivalent circuit simulation under a linear region ($V_{gs}$ = 0.65 V, $V_{ds}$ = 0.05 V) and saturation region ($V_{gs}$ = 0.65 V, $V_{ds}$ = 0.65 V) is depicted in Figure 9. Excellent agreement is obtained over the entire frequency. The maximum error in the linear region and saturation region are 0.2504% and 2.689%, which confirms the accuracy of the proposed extraction method.

**Table 1.** The fitted constant $N_{i0}$ and the corresponding confidence interval for 3 nm GAA nanosheet under $V_{gs}$ = −0.65 V, $V_{ds}$ = 0 V.

| Parameters | Values | Confidence Intervals (CIs) |
|---|---|---|
| $N_{10}$ ($1 \times 10^{-31}$) | 3.747 | (3.745, 3.749) |
| $N_{20}$ ($1 \times 10^{-31}$) | 1.995 | (1.991, 1.998) |
| $N_{30}$ ($1 \times 10^{-17}$) | 1.466 | (1.465, 1.467) |
| $N_{40}$ ($1 \times 10^{-31}$) | 2.185 | (2.179, 2.182) |
| $N_{50}$ ($1 \times 10^{-17}$) | 1.565 | (1.564, 1.566) |

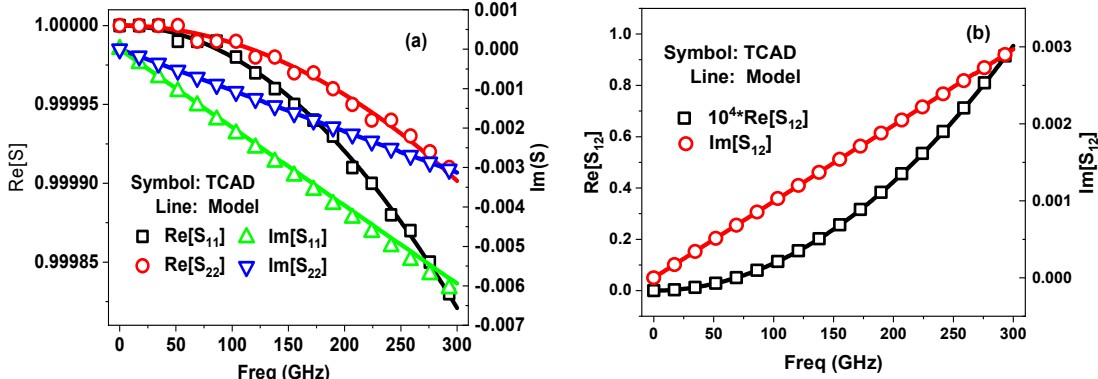

**Figure 8.** Comparison between simulated (symbols) and modeled S-parameters (solid lines) under OFF-state bias $V_{gs}$ = −0.65 V, $V_{ds}$ = 0 V. (**a**) S11 and S22, (**b**) S12.

**Table 2.** The fitted constant $K_{i0}$ and the corresponding confidence interval under $V_{gs}$ = 0.65 V, $V_{ds}$ = 0.05 V.

| Parameters | Values | Confidence Intervals (CIs) |
|---|---|---|
| $K_{10}$ ($1 \times 10^{-16}$) | 0.2968 | (0.2968, 0.2968) |
| $K_{20}$ ($1 \times 10^{-31}$) | 8.072 | (8.07, 8.075) |
| $K_{30}$ ($1 \times 10^{-16}$) | 0.2772 | (0.2772, 0.2773) |
| $K_{40}$ ($1 \times 10^{-31}$) | 10.23 | (10.23, 10.24) |
| $K_{50}$ ($1 \times 10^{-18}$) | −0.8724 | (−0.8726, −0.8721) |
| $K_{60}$ ($1 \times 10^{-4}$) | 0.2706 | (0.2705, 0.2706) |
| $K_{70}$ ($1 \times 10^{-17}$) | −1.187 | (−1.188, −1.187) |
| $K_{80}$ ($1 \times 10^{-3}$) | 2.357 | (2.357, 2.357) |
| $K_{90}$ ($1 \times 10^{-23}$) | 0.2484 | (0.2469, 0.2498) |

Figure 10a shows the comparison between simulated and modeled maximum available gain (*MAG*), maximum stable gain (*MSG*), Mason's unilateral gain (*MUG*), unilateral figure of merit (*Uf*) and $H_{21}$. As expected, ADS-modeled data agree well with the Sentaurus simulated one over the whole frequency range. The extracted cutoff frequency $F_t$ and maximum oscillation frequency $F_{max}$ under different biases shown in Figure 10b. As expected, $F_t$ and $F_{max}$ increase first and then decrease with increasing $V_{gs}$. Excellent agreement between the simulated and modeled RF figures of merit (FOM) proves that the extracted small-signal equivalent circuit can effectively predict the RF response of the device up to 300 GHz. Subsequently, the bias dependence of intrinsic model parameters is discussed. Figure 11 shows the variation of extracted critical model parameters as a function of $V_{gs}$. $C_{gsi}/C_{gdi}$ gradually increases as $V_{gs}$ increases, and then becomes flat when $V_{gs}$ is large enough, which is consistent with the typical C-V curve of MOS capacitor.

Moreover, $C_{gdi}$ is smaller than $C_{gsi}$ under the same $V_{ds}$, and the difference between two capacitances increases with increasing $V_{ds}$, especially in the saturation region, which is the result of the thinner depletion layer near the drain region. $g_m$ sharply increases when the transistor operates in the saturation region, and the increasing trend slows down once the transistor enters the linear region, due to the carrier velocity saturation effect under high electric field. $\tau_m$ is found to first increase and then decrease with increasing $V_{gs}$. Similar to the planar MOSFET, $\tau_m$ can be expressed as:

$$\tau_m \propto \frac{L_G^2}{\mu_n \left( V_{gs} - V_{th} \right)} \tag{40}$$

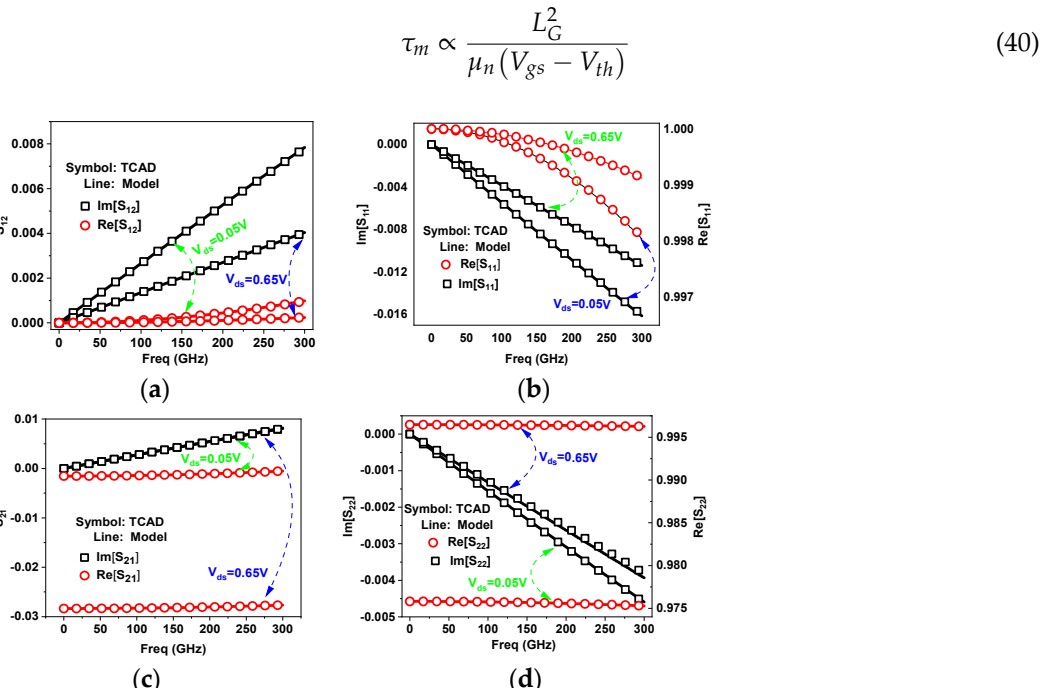

**Figure 9.** Comparison between TCAD-simulated and -modeled S-parameters under $V_{gs}$ = 0.65 V, $V_{ds}$ = 0.05 V and $V_{gs}$ = 0.65 V, $V_{ds}$ = 0.65 V. (**a**) S12, (**b**) S11, (**c**) S21 and (**d**) S22.

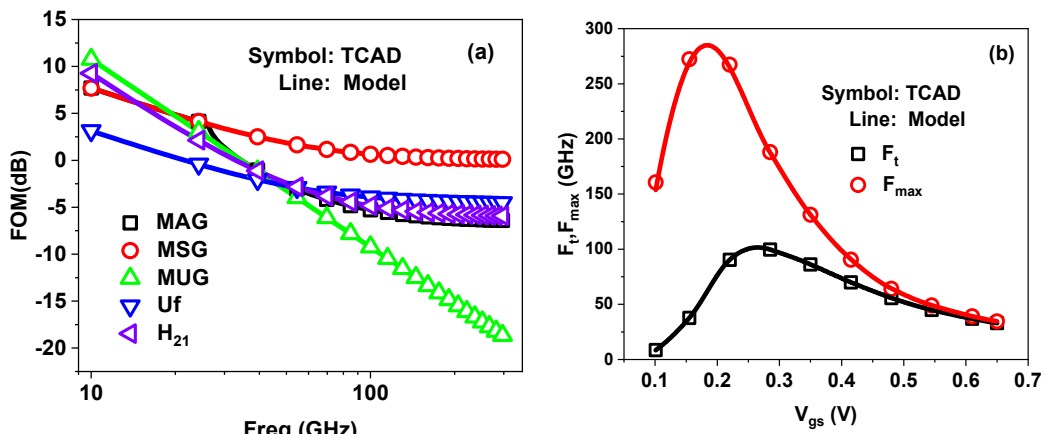

**Figure 10.** Comparison between simulated (symbols) and modeled (solid lines) (**a**) *MAG, MSG, MUG, Uf* and *H₂₁* under $V_{gs}$ = 0.65 V, $V_{ds}$ = 0.65 V. (**b**) $F_t$ and $F_{max}$ as a function of Vgs under Vds = 0.05 V.

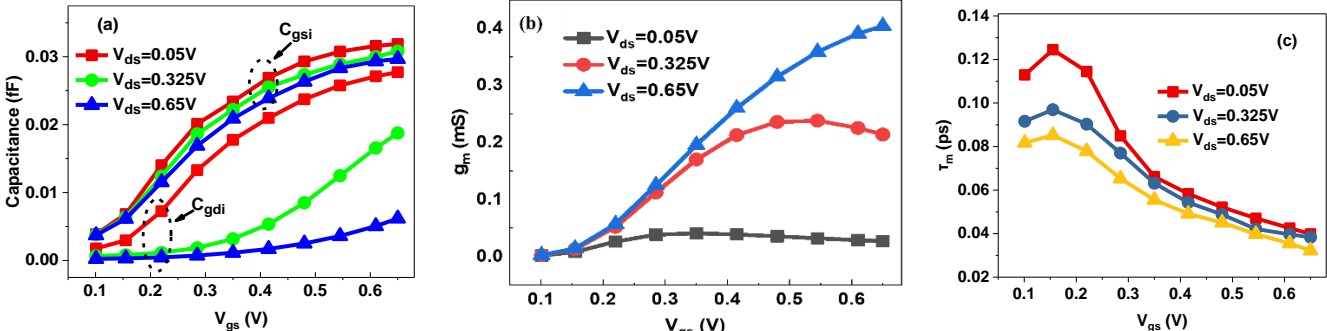

**Figure 11.** Variation of extracted model parameters as a function of Vgs, (**a**) $C_{gsi}$ and $C_{gdi}$, (**b**) $g_m$ and (**c**) $\tau_m$.

In the above formula, the mobility $\mu_n$ decreases with increasing $V_{gs}$. These two items in the denominator contribute a competitive relationship, which causes $\tau_m$ to increase first and then decrease.

## 4. Conclusions

In this paper, an improved RF small-signal equivalent circuit model and corresponding parameter extraction method is proposed for GAAFET at 3 nm node. The equivalent circuit contains both the extrinsic GSG test layout and intrinsic device. The parasitic parameters of the GSG test layout are first obtained by an improved five-step method. The model parameters of the intrinsic GAAFET are analytically extracted with the help of nonlinear rational function fitting. The validity of the proposed small-signal equivalent circuit model and the corresponding extraction method have been demonstrated, through excellent agreement between TCAD/HFSS 3D full-wave simulation and equivalent circuit frequency response up to 300 GHz. The error is lower than 2.69% for intrinsic devices, and lower than 0.89% for the extrinsic GSG test layout in the whole frequency range. Moreover, an obvious influence of WFV is found on the variation of critical model parameters, especially under the condition of smaller gate grain size.

**Author Contributions:** Conceptualization and methodology Z.L.; validation X.L.; writing—original draft preparation, H.G.; writing—review and editing, Y.S. (Yabin Sun); visualization, Y.L.; supervision, Y.S. (Yanling Shi); formal analysis and data curation, S.H. All authors have read and agreed to the published version of the manuscript.

**Funding:** This research was funded by the National Natural Science Foundation of China, grant number 61974056; Science Foundation of Shanghai, grant number 19ZR1471300; Shanghai Science and Technology Innovation Action Plan, grant number 19511131900; and Shanghai Science and Technology Explorer Plan, grant number 21TS1401700.

**Data Availability Statement:** The datasets generated during and/or analysis during the current study are available from the corresponding author on reasonable request.

**Conflicts of Interest:** The authors declare no conflict of interest.

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
