# Peer review of "Sub-THz Small-Signal Equivalent Circuit Model and Parameter Extraction for 3 nm Gate-All-Around Nanosheet Transistor"

_processes, doi:10.3390/pr10061198_

Round 1

Reviewer 1 Report

In this paper the authors present a model for a novel method that can extract sub-THz circuit parameters for nm-size FETs from several steps. The results has been compared with simulation and gave a good agreement. The paper is clearly written and I feel it is publishable after minor corrections listed below are made.

  1. There are too many acronyms that make it a little bit difficult to read through the paper. Some of them can be explicitly repeated, or can be summarized in a table.
  2. The symbols for the electrodes in Figure 4 is somehow ambiguous. The symbols should be consistent among the picture, circuit and maintext.
  3. It was impossible to understand how G_gde and C_gde can be determined, because such a test configuration is not made in Figure 4. Is it extracted from the overall device performance? Maybe this point needs to be clarified, although I might have overlooked it.

Author Response

Thanks for the reviewer's comment. The response to the comment has been attached in the file named "response to reviewer 1 comment"

Reviewer 2 Report

The manuscript by Yabin Sun et al. reports a study of Sub-Thz small-sgnial equivalent circuit model and parameter extraction for 3 nm gate-all-around nanosheet transistor. And they obtained great parameter through simulation and modeling. Authors have well organized and written this work. Therefore, I support the publication of this manuscript on processes. However, there are some minor requirements.

  1. About Figure 1, author should better describe the structure of the device by layer in manuscript and figure.
  2. Simulation method and how to get those results must be described. It is not clear how the simulation is performed. The authors need to write more detail about the simulation method.

Author Response

Thanks for the reviewer's comment. The response to the comment has been attached in the file named "response to reviewer 2 comment"
